# Polyphenols Extracted from *Artemisia annua* L. Exhibit Anti-Cancer Effects on Radio-Resistant MDA-MB-231 Human Breast Cancer Cells by Suppressing Stem Cell Phenotype, β-Catenin, and MMP-9

**DOI:** 10.3390/molecules25081916

**Published:** 2020-04-21

**Authors:** Young Shin Ko, Eun Joo Jung, Se-il Go, Bae Kwon Jeong, Gon Sup Kim, Jin-Myung Jung, Soon Chan Hong, Choong Won Kim, Hye Jung Kim, Won Sup Lee

**Affiliations:** 1Department of Pharmacology, College of Medicine, Institute of Health Sciences, Gyeongsang National University, 816 Beongil 15 Jinjudaero, Jinju 52727, Korea; shini33@naver.com; 2Departments of Biochemistry, Institute of Health Sciences, Gyeongsang National University College of Medicine, 90 Chilam-dong, Jinju 660-702, Korea; eunjoojung@gnu.ac.kr (E.J.J.); cwkim3540@hanmail.net (C.W.K.); 3Departments of Internal Medicine, Institute of Health Sciences and Gyeongsang National University Hospital, Gyeongsang National University College of Medicine, 90 Chilam-dong, Jinju 660-702, Korea; gose1@hanmail.net; 4Departments of Radiation Oncology, Institute of Health Sciences and Gyeongsang National University Hospital, Gyeongsang National University College of Medicine, 90 Chilam-dong, Jinju 660-702, Korea; blue129j@hanmail.net; 5School of Veterinary and Institute of Life Science, Gyeongsang National University, 900 Gajwadong, Jinju 660-701, Korea; gonskim@gnu.ac.kr; 6Departments of Neurosurgery, Institute of Health Sciences and Gyeongsang National University Hospital, Gyeongsang National University College of Medicine, 90 Chilam-dong, Jinju 660-702, Korea; gnuhjjm@gnu.ac.kr; 7Departments of Surgery, Institute of Health Sciences and Gyeongsang National University Hospital, Gyeongsang National University College of Medicine, 90 Chilam-dong, Jinju 660-702, Korea; hongsc@gnu.ac.kr

**Keywords:** breast cancer cells, polyphenols, *Artemisia annua* L., stem cells, EMT

## Abstract

*Artemisia annua* L. has been reported to show anti-cancer activities. Here, we determined whether polyphenols extracted from *Artemisia annua* L. (pKAL) exhibit anti-cancer effects on radio-resistant MDA-MB-231 human breast cancer cells (RT-R-MDA-MB-231 cells), and further explored their molecular mechanisms. Cell viability assay and colony-forming assay revealed that pKAL inhibited cell proliferation on both parental and RT-R-MDA-MB-231 cells in a dose-dependent manner. The anti-proliferative effects of pKAL on RT-R-MDA-MB-231 cells were superior or similar to those on parental ones. Western blot analysis revealed that expressions of cluster of differentiation 44 (CD44) and Oct 3/4, matrix metalloproteinase-9 (MMP-9) and signal transducer and activator of transcription-3 (STAT-3) phosphorylation were significantly increased in RT-R-MDA-MB-231 cells compared to parental ones, suggesting that these proteins could be associated with RT resistance. pKAL inhibited the expression of CD44 and Oct 3/4 (CSC markers), and β-catenin and MMP-9 as well as STAT-3 phosphorylation of RT-R-MDA-MB-231. Regarding upstream signaling, the JNK or JAK2 inhibitor could inhibit STAT-3 activation in RT-R-MDA-MB-231 cells, but not augmented pKAL-induced anti-cancer effects. These findings suggest that c-Jun N-terminal kinase (JNK) or Janus kinase 2 (JAK2)/STAT3 signaling are not closely related to the anti-cancer effects of pKAL. In conclusion, this study suggests that pKAL exhibit anti-cancer effects on RT-R-MDA-MB-231 cells by suppressing CD44 and Oct 3/4, β-catenin and MMP-9, which appeared to be linked to RT resistance of RT-R-MDA-MB-231 cells.

## 1. Introduction

In recent decades, phytochemicals have been given much attention as potential candidates for cancer treatment because they exhibit anti-cancer effects without any noticeable toxicities [1]. Among phytochemicals, natural polyphenols are abundantly present in various edible fruits, vegetables and herbs, which are assumed to be related to a reduction in cancer risk [2,3]. *Artemisia annua* L., (Gaddongsook, Korean), an annual herb, has been used for a long time as a Korean folk medicine for the treatment of malaria, fever, and neurologic disorders [4,5]. In addition, it possesses anti-cancer activity [6]. However, the molecular mechanisms for the anti-cancer activities of Korean *Artemisia annua* still need elucidating.

Breast cancer is considered as one of the leading causes of cancer-related death worldwide, and its incidence is increasing in Korea [7,8]. Although the treatment outcomes for breast cancer have been improved, resistance to radiation (RT) and/or chemotherapy (CT) is a big obstacle to curing cancer. One of the major causes for the resistance to RT or CT is cancer stem cells (CSCs). Therefore, the development of a certain therapy targeted at CSCs holds hope for curing cancer.

Our team previously established radio-resistant MDA-MB 231 human breast cancer cells (RT-R-MDA-MB 231 cells) which exhibit enhanced aggressiveness, and cancer stem cell features [9,10]. These cells also manifest epithelial–mesenchymal transition (EMT), a process by which epithelial cells gain migratory and invasive properties to become mesenchymal stem cells. This means that the induction of EMT could change non-CSCs into CSCs [11,12,13]. From this evidence, EMT is also considered as a mechanism for the resistance to RT or CT [14]. Therefore, EMT and CSCs could be good targets to overcome the resistance to RT or CT.

We previously demonstrated that polyphenols extracted from Korean *A. annua* L. (pKAL) exhibited anti-cancer effects by inhibiting the EMT process without showing any significant cytotoxicity on normal cells [15,16]. Therefore, we hypothesized that pKAL harbors anti-cancer properties in overcoming radio resistance (RT-resistance) by suppressing CSCs and EMT. If pKAL exhibit significant anti-cancer effects on RT-R-MDA-MB-231 cells, pKAL-based phytotherapy will be an applicable and helpful option against resistance to RT or CT in breast cancer. In this study, we established RT-R-MDA-MB-231 cells following the previous protocol [9], determined whether pKAL would exhibit anti-cancer effects on the RT-R breast cancer cells, and further explored their molecular mechanisms by assessing the effects of pKAL on expressions of the proteins that were significantly higher expressed in RT-R-MDA-MB-231 cells than parental MDA-MB-231 cells, and assumed to be related to RT-resistance.

## 2. Results

### 2.1. pKAL Inhibited Growth of RT-R-MDA-MB-231 Cells, and Its Efficacy Was Superior or Similar to that on Parental MDA-MB-231 Cells

To investigate the anti-cancer activity of pKAL on RT-R-MDA-MB-231 cells, we treated them with indicated concentrations (up to 100 μg/mL) of pKAL for 72 h. MTT assay revealed that pKAL inhibited the growth of RT-R-MDA-MB-231 cells in a dose-dependent manner, and that RT-R-MDA-MB-231 cells were as sensitive to pKAL as parental MDA-MB-231 cells during 72 h-pKAL treatment (Figure 1A). In a colony-forming assay, RT-R-MDA-MB-231 cells grew far faster than parental MDA-MB-231 cells (Figure 1B). The anti-cancer activity of pKAL on RT-R-MDA-MB-231 cells was superior or similar to that of parental MDA-MB-231 cells. These findings suggest that pKAL might harbor anti-cancer effects on RT-R human breast cancer cells, and its efficacy was superior or similar to that on parental MDA-MB-231 cells.

### 2.2. pKAL Significantly Inhibited Expression of Stem Cell Markers (CD44, And Oct 3/4), Β-Catenin, and MMP-9 that Were Overexpressed in RT-R-MDA-MB-231 Cells Compared to Parental MDA-MB-231 Cells

In this study, several markers-CD44 and Oct-3/4 (octamer-binding transcription factor 3/4), β-catenin, and MMP-9-were chosen for RT resistance, for they were significantly increased after acquiring resistance to RT in previous study [9]. In addition, they are related to CSCs, cancer progression, and EMT. CD44 and Oct-3/4 are the most robust surface markers for CSCs [17,18], and these are strongly linked to RT resistance [19,20,21]. WNT/beta-catenin mediates the radiation resistance of breast progenitor cells [14]. β-catenin is an important molecule involved in EMT [22,23,24]. MMP-9 is most widely associated with cancer progression, due to its role in extracellular matrix remodeling and angiogenesis [25,26]. Therefore, we investigated the effects of pKAL on the expression of CD44, Oct 3/4, β-catenin, and MMP-9 of RT-R-MDA-MB-231 cells as well as of parental MDA-MB-231 cells. Western blot analysis revealed that while the expression of β-catenin was slightly increased in RT-R-MDA-MB-231 compared to those in parental MDA-MB-231 cells, the expressions of CD44, Oct 3/4, and MMP-9 were significantly increased in RT-R-MDA-MB-231 cells, suggesting that the expressions of CD44, Oct 3/4, and MMP-9 would be related to RT-resistance of MDA-MB-231 cells. Even though the difference in the expression of β-catenin was not statistically significant between parental and RT-R-MDA-MB-231 cells, pKAL inhibited the expression of β-catenin in a dose-dependent manner in RT-R-MDA-MB-231 cells, and its effect was significant at the concentration of 50 μg/mL pKAL or higher (Figure 2A–D, respectively). These findings suggest that the pKAL have the ability to inhibit the proteins that appeared to be related to RT resistance and associated with the stemness characteristics, and EMT phenotype of RT-R-MDA-MB-231 cells.

### 2.3. pKAL Inhibited STAT 3 Activity that Compared to in Parental MDA-MB-231 Cells, Was Significantly Increased in RT-R-MDA-MB-231 Cells

It was reported that stem cell-like breast cancer cells showed high STAT3 activation and the JAK2/STAT3 signaling pathway is important in maintaining stem cell characteristics [27]. Upregulated STAT3 activity also participates and plays an important role in EMT [28,29]. Here, we investigated the effects of pKAL on STAT3 activity in RT-R-MDA-MB-231 cells, as well as parental MDA-MB-231 cells. Western blot analysis revealed that STAT3 activity in RT-R-MDA-MB-231 cells was significantly increased compared to that in parental MDA-MB-231 cells, and that pKAL inhibited the STAT3 activity of RT-R-MDA-MB-231 cells in a dose-dependent manner (Figure 3). These findings suggest that pKAL might show anti-cancer effects on RT-R-MDA-MB-231 cells by suppressing STAT3 activity, which appeared to be linked to stemness characteristics and EMT phenotype as well as RT-resistance.

### 2.4. The Anti-Cancer Effects of pKAL on RT-R-MDA-MB-231 Cells Were not Closely Associated with JAK2/STAT3 Signaling

As previously mentioned, JAK2/STAT3 signaling pathway is reportedly important in maintaining stem cell characteristic [27]. To confirm that, here we performed a JNK or JAK2 inhibitor test by assessing the anti-cancer effects of pKAL on CD44, Oct 3/4, β-catenin, and MMP-9 as well as the STAT 3 activity of RT-R-MDA-MB-231 cells. The reason why we add a JNK inhibitor is because JNK signaling is also related to STAT3 activity [30], and is to confirm that the JAK2/STAT3 signaling pathway uniquely plays an important role in maintaining self-renewal and tumor initiating capacity of CSCs, and enhancing stemness characters. Western blot analysis revealed that pKAL inhibited STAT3 activity, which is up-regulated in RT-R-MDA-MB-231 cells compared to in parental MDA-MB-231 cells, (Figure 4A,B), and that the JNK or JAK2 inhibitor alone also inhibited STAT3 activity on both RT-R-MDA-MB-231 cells and MDA-MB-231 cells (Figure 4A,B). However, the JNK or JAK2 inhibitor could neither augment the anti-cancer effects of pKAL on STAT3 activity (Figure 4A,B), nor on the expressions of CD44, Oct 3/4, β-catenin, and MMP-9 of RT-R-MDA-MB-231 cells (Figure 5). Moreover, the JNK or JAK2 inhibitor could not suppress the expression CD44 of RT-R-MDA-MB-231 cells (Figure 5A,B), while the JAK2 inhibitor could inhibit the expressions of Oct 3/4, β-catenin, and MMP-9 in RT-R-MDA-MB-231 cells (Figure 5C,D). These findings suggest that pKAL-induced anti-cancer effects might not be closely associated with JAK2/STAT3 signaling, while JAK2 inhibited STAT3 activity and the expressions of Oct 3/4, β-catenin, and MMP-9 of RT-R-MDA-MB-231 cells.

## 3. Discussion

The present study was designed to determine whether pKAL could overcome the RT resistance of RT-R-MDA-MB-231 cells by showing its anti-cancer activities on RT-R-MDA-MB-231 human breast cancer cells, and to further explore their molecular mechanisms, focusing on changes in the expression of proteins that might be related to the RT resistance of RT-R-MDA-MB-231 cells and are used as markers for CSCs, EMT, and cancer progression. In this study, pKAL clearly demonstrated a significant anti-cancer effect on RT-R-MDA-MB-231 cells, and its efficacy was superior or similar to that of parental MDA-MB-231 cells. In addition, pKAL inhibited expressions of the proteins (CD44, Oct 3/4, β-catenin and MMP-9) that are assumed to be related to the RT resistance of RT-R-MDA-MB-231 human breast cancer cells [9]. With these findings, we concluded that pKAL exhibits anti-cancer effects on RT-R-MDA-MB-231 cells, by suppressing the expressions of proteins which appeared to be linked to the RT resistance, CSCs, EMT, and cancer progression of RT-R-MDA-MB-231 cells. Regarding this conclusion, readers may raise some questions. The first one would be about the relationship between RT resistance and stemness. Another would be about the validity of CD44 and Oct 3/4 as the stem cell markers. Since the two questions are very closely related, we here discuss them together. Regarding a link between RT resistance and stemness, this question still has some controversies, but it is becoming apparent that cancer cells showing RT resistance are enriched with the cells showing stem cell and EMT phenotypes [31]. In addition, CD44 is frequently used as a stem cell marker and strongly linked to RT resistance [19,21]. Oct-3/4 also participates in the self-renewal of undifferentiated stem cells [18] and RT resistance [20]. Therefore, many investigators believe that EMT and cancer stemness are the main mechanisms for RT resistance [14,32,33].

The third one would be about whether β-catenin and MMP-9 are involved in RT resistance. To clearly answer this question, further study is needed, but β-catenin and MMP-9 are important molecules clearly involved in EMT [22,23,24,34] and cancer progression [25,26,35]. In addition, these proteins were significantly increased in RT-R MDA-MB-231 cells.

The fourth one would be whether STAT3 activity is involved in RT resistance, and controlling the expressions of CD44, Oct 3/4, β-catenin and MMP-9. With our results, we cannot give a clear answer yet, because STAT3 signaling is highly inter-connected with other signals while participating in breast cancer progression, EMT, and the maintenance of CSC characteristics. In addition, other molecular signaling pathways such as WNT and NOTCH signaling pathways are also involved in EMT and the acquisition of stem cell properties [36,37]. Even though we could not give clear evidence, it is still possible that STAT 3 activity is at least partly involved in RT resistance of MDA-MB-231 breast cancer cells and the regulation of the expressions of CD44, Oct 3/4, β-catenin and MMP-9 because many reports showed that STAT3 activity is closely associated with stem cell-like traits, EMT, and drug resistance [27,38].

The fifth one would be whether stem cell and EMT phenotypes are related to cancer progression and the aggressive phenotypes that are observed in RT-R-MDA-MB-231 cells, compared to parental MDA-MB-231 cells. This feature should be related to radiation resistance or radiation itself, because populations of the irradiated CSCs become more aggressive than those of non-irradiated CSCs [31]. Actually, MDA-MB-231 cells that belong to basal-like cancer with a high population of CSCs grows more slowly than luminal type MCF-7 breast cancer cells with a low population of CSCs (data not shown) [27]. This finding suggests that the aggressive feature could be attributed to RT resistance rather than that of the CSCs, themselves. In addition, highly proliferative features of RT-R-MDA-MB-231 cells were observed after acquiring RT resistance [9].

The weakness of this study is that the main mechanism of overcoming the drug resistance of pKAL was not clear about how pKAL inhibited the expression of CD44, Oct 3/4 β-catenin and MMP-9 that might be associated with the inhibition of RT resistance in RT-R-MDA-MB-231 cells. This would be a very important question. Until now, many therapies that target the EMT/CSC phenotype have been identified, but the exact mechanisms are still unclear [39,40]. Studies suggest that the cancer progression and metastasis is associated with the acquisition of stemness and EMT pattern. In addition, several other studies have suggested that the two functions (acquisition of stemness and EMT pattern) are closely linked in cancer progression [13,36]. Regarding the relationship between radiation resistance and the acquisition of stemness and EMT pattern in RT-R-MDA-MB-231 cells, further investigations are required.

Lastly, the merit of pKAL is that, as previously mentioned, pKAL exhibits anti-cancer effects without any noticeable toxicities. At the concentrations where pKAL showed no toxicity on the normal cells [16], it induced anti-cancer activity on RT-R-MDA-MB-231 cells with similar efficacy to paternal MDA-MB-231 cells. This finding suggests that pKAL could be clinically applicable and helpful for RT-resistant end-stage breast cancer, especially for the patients with a poor general condition who cannot tolerate other palliative conventional chemotherapy.

In conclusion, this study suggests pKAL exhibits anti-cancer effects on RT-R-MDA-MB-231 cells, by suppressing CD44 and Oct 3/4, β-catenin and MMP-9 which appeared to be linked to the RT resistance of RT-R-MDA-MB-231 cells. The inhibition of these four proteins by pKAL was not associated with JNK-, JAK-2-associated STAT3 activity, while pKAL inhibited STAT3 activity in RT-R-MDA-MB-231 cells. This study provides evidence that pKAL might have an anti-cancer property on RT-R human breast cancer cells, and can be used as therapeutic potential for the treatment of breast cancer.

## 4. Materials and Methods

### 4.1. Preparation of Polyphenols from Korean *Artemisia annua* L.

Polyphenols were extracted from Korean *A. annua* L. (pKAL) and characterized by Professor Shin (Gyeongsang National University, Jinju, Korea) [5]. Briefly, the lyophilized Korean *A. annua* L. (KAL) tissues including roots, stems, leaves, and flowers (10 g) were ground into powder, extracted in ethyl acetate (300 mL) at 80 °C for 20 h, and eluted with a mixture of methanol:dichloromethane (1:5, 25 mL). The isolated polyphenol mixtures were identified by HPLC-MS/MS according to the previous method [41].

### 4.2. Chemicals and Reagents

The MDA-MB-231 human breast cancer cells that were obtained from the Korea Cell Line Bank (Seoul, Korea), were cultured in RPMI 1640 medium from HyClone (Logan, UT, USA) supplemented with 10% (v/v) fetal bovine serum (FBS) from GIBCO BRL (Grand Island, NY, USA), 1 mM L-glutamine, 100 U/mL penicillin, and 100 μg/mL streptomycin at 37 °C in a humidified atmosphere of 95% air and 5% CO_2_. Antibodies against anti-octamer-binding transcription factor (Oct3/4) and β-catenin were purchased from Santa Cruz Biotechnology (Dallas, Texas, TX, USA). Antibodies against CD44, STAT3, and phosphor-STAT3 were purchased from Cell Signaling Technology (Beverly, MA, USA). An antibody against β-actin was from Sigma (Beverly, MA, USA). Peroxidase-labeled goat anti-rabbit was purchased from Santa Cruz Biotechnology and an enhanced chemiluminescence (ECL) kit was purchased from Bio-Rad (Hercules, CA, USA). All other chemicals not specifically cited here were purchased from Sigma-Aldrich (St. Louis, MO, USA).

### 4.3. Establishment of Radio-Resistant MDA-MB-231 Human Breast Cancer Cells (RT-R-MDA-MB-231 Cells)

According to the previous protocol [9], RT-R-MDA-MB-231 cells were generated by applying repetitive small doses of X-ray irradiation (2 Gy) using a 6-MV photon beam produced by a linear accelerator (Clinac 21EX, Varian Medical Systems, Inc., Palo Alto, CA) until a final dose of 50 Gy was achieved. The radiation dose rate was 1.0 Gy/min, and the cell medium was changed immediately after irradiation. When the cells reached ~90% confluence, they were sub-cultured into new flasks. At about 70% confluence, irradiation was resumed.

### 4.4. Cell Viability Assay

The cell viability assay was performed with 3-(4, 5-dimethylthiazol-2-yl)-2, 5-diphenyltetrazolium bromide (MTT) assay. For the MTT assay, paternal and RT-R-MDA-MB-231 cells were treated with pKAL for 24–72 h, and then incubated in 0.1 mg/mL MTT solution for 3 h at 37 °C in the dark. The absorbance of each well was measured at 540 nm with an enzyme-linked immunosorbent assay (ELISA) reader (Sunnyvale, CA, USA).

### 4.5. Colony Formation Assay

Parental or RT-R-MDA-MB-231 cells were seeded in 6-well plates (1 × 10^3^ cells/well). Then, the cells were treated with pKAL at the indicated concentrations at 37 °C. After 24 h of treatment, the culture medium was discarded and replaced with fresh complete medium every 2–3 days. After 10 days, the medium was discarded, and each well was washed with PBS. The colonies were fixed in 100% methanol for 10 min, and then stained with 0.1% Giemsa staining solution for 30 min at room temperature. The number of visible colonies was counted.

### 4.6. Western Blot Analysis

After being treated with pKAL at the indicated concentrations for 24 h, the cells were harvested and lysed. Their proteins were quantified by the Bradford method. The proteins of the extracts were resolved by electrophoresis, electrotransferred to a polyvinylidene difluoride membrane (Amersham Biosciences, Little Chalfont, UK), and then the membrane was incubated with the primary antibodies followed by a conjugated secondary antibody to peroxidase. The blots were developed under an ECL detection system (Bio-Rad).

### 4.7. Gelatin Aymography

The gelatinolytic activities for MMP-9 (gelatinase-B) were assessed as previously described [42]. Briefly, polyacrylamide gels containing 1 mg/mL gelatin were run at 120 V, washed in 2.5% Triton X-100 for 1 h, and then incubated for 16 h at 37 °C in an activation buffer (50 mM Tris-HCl, 20 mM NaCl, 5 mM CaCl_2_ and 0.02% Brij35, pH 7.5). After staining with Coomassie Blue for 2 h, the gel was washed with a solution of 10% glacial acetic acid and 30% methanol for 1 h. White lysis zones indicating gelatin degradation were revealed by staining with Coomassie Brilliant Blue.

### 4.8. Statistical Analysis

The results were expressed as means ± SEM from at least three independent experiments. Significant differences were determined by the one-way analysis of variance (ANOVA) with post-test Newman–Keuls for comparison of at least three treatment groups and Student’s *t*-test for two groups. Statistical significance was defined as *p* < 0.05.

## Figures and Tables

**Figure 1 molecules-25-01916-f001:**
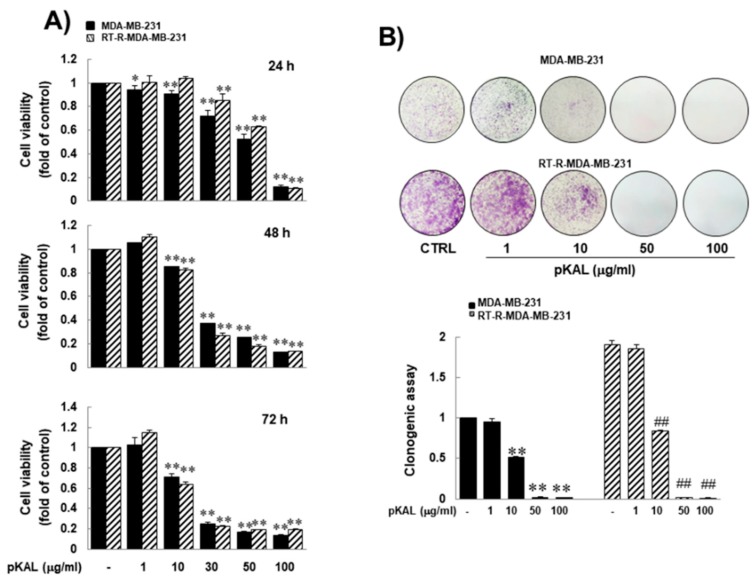
Similar inhibitory effect of pKAL on RT-R-MDA-MB-231 human breast cancer cells, which was similar to that on parental MDA-MB-231 cells. (**A**) Parental and RT-R-MDA-MB-231 cells were treated with the indicated concentrations of pKAL for 24–72 h, and then cell viability was performed by MTT assay. (**B**) Parental and RT-R-MDA-MB-231 cells were treated with the indicated concentrations of pKAL for 24 h. Then, the medium was changed with fresh complete medium. Ten days later, colony formation assay was performed as described in the methods. They were quantified by counting the colonies. The values are expressed as the means ± SEM from three independent experiments (* *p* < 0.05 vs. each control, ** *p* < 0.01 vs. each control, and ^##^
*p* < 0.01 vs. control for RT-R-MDA-MB-231 cells).

**Figure 2 molecules-25-01916-f002:**
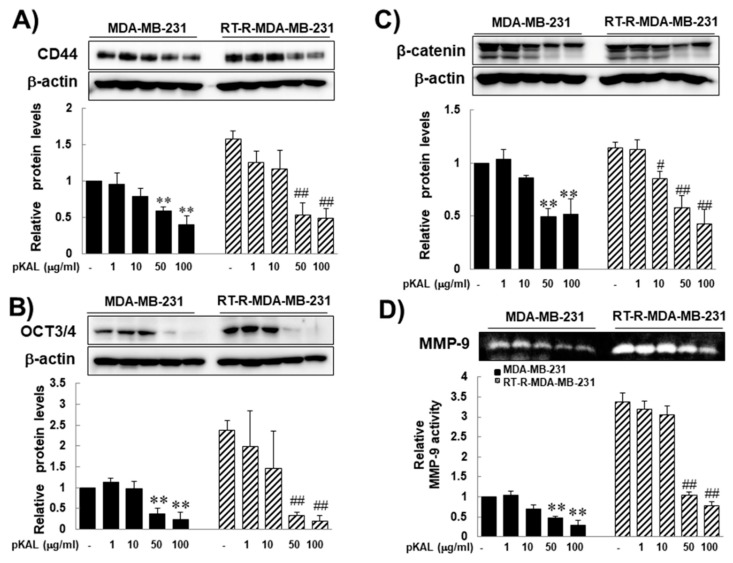
Inhibitory effects of *Artemisia annua* L. (pKAL) on the expression of stem cell markers (CD44 and Oct 3/4), β-catenin and the activity of MMP-9 in both parental and RT-R-MDA-MB-231 cells. The cells were treated with the indicated concentrations of pKAL for 24 h. The expressions of CD44 (**A**), Oct3/4 (**B**) and β-catenin (**C**) were measured from the cell lysates by Western blot analysis. (**D**) The activity of MMP-9 was examined by gelatin zymography. The band density was quantified by densitometry, and the values are expressed as the means ± SEM from three independent determinations (** *p* < 0.01 vs. control for parental MDA-MB-231 cells, ^#^
*p* < 0.05 vs. control for RT-R-MDA-MB-231 cells, and ^##^
*p* < 0.01 vs. control for RT-R-MDA-MB-231 cells).

**Figure 3 molecules-25-01916-f003:**
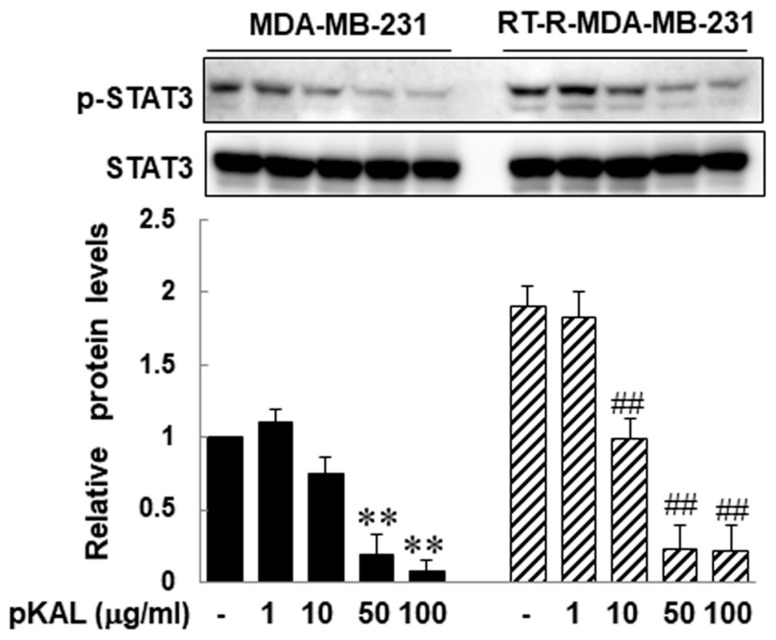
Inhibitory effects of pKAL on STAT3 phosphorylation of both parental and RT-R-MDA-MB-231 cells. The cells were treated with indicated concentrations of pKAL for 24 h. The expressions of *p*-STAT3 and STAT3 proteins were measured by western blot analysis. The band density was measured by densitometry, and the values are expressed as the means ± SEM from three independent determinations (** *p* < 0.01 vs. control for parental MDA-MB-231 cells, and ^##^
*p* < 0.01 vs. control for RT-R-MDA-MB-231 cells).

**Figure 4 molecules-25-01916-f004:**
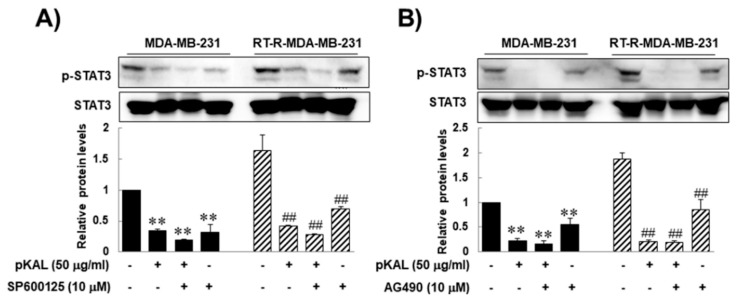
The effects of the JNK or JAK2 inhibitor alone or in combination with pKAL on the STAT3 phosphorylation of both parental and RT-R-MDA-MB-231 cells. The cells were treated with the indicated agent alone or in combination with pKAL for 24 h. The expressions of p-STAT3 or STAT3 were measured by Western blot analysis and densitometry, and the values are expressed as the means ± SEM from three independent determinations (** *p* < 0.01 vs. control for parental MDA-MB-231 cells, and ^##^
*p* < 0.01 vs. control for RT-R-MDA-MB-231 cells).

**Figure 5 molecules-25-01916-f005:**
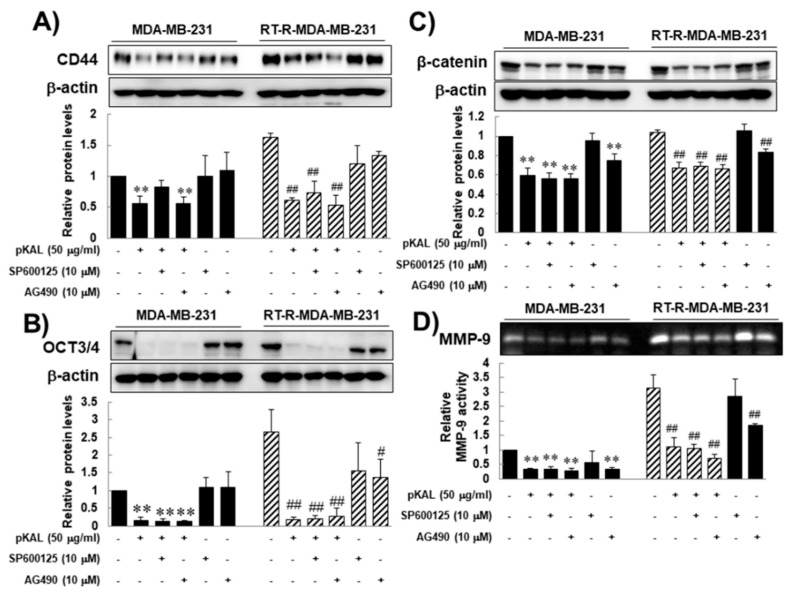
Inhibitory effects of pKAL in combination with the JNK or JAK2 inhibitor on the expression of stem cell markers (CD44 and Oct 3/4), β-catenin, and the activity of MMP-9 in both parental and RT-R-MDA-MB-231 cells. The cells were treated with indicated concentrations of pKAL for the indicated concentrations of pKAL for 24 h. The expressions of CD44 (A), Oct3/4 (B) and β-catenin (C), and the activity of MMP-9 (D) were measured as described in the Figure 2 legend. The values are expressed as the means ± SEM from three independent determinations (** *p* < 0.01 vs. control for parental MDA-MB-231 cells, ^#^
*p* < 0.05 vs. control for RT-R-MDA-MB-231 cells, and ^##^
*p* < 0.01 vs. control for RT-R-MDA-MB-231 cells).

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
