# Peer review of "Polyphenols Extracted from Artemisia annua L. Exhibit Anti-Cancer Effects on Radio-Resistant MDA-MB-231 Human Breast Cancer Cells by Suppressing Stem Cell Phenotype, β-Catenin, and MMP-9"

_molecules, 2020, doi:10.3390/molecules25081916_

Round 1

Reviewer 1 Report

The submitted manuscript on Polyphenols from Artemisia annua L. gives a good example of quality manuscript. Nevertheless, there are several points to be corrected or improved before the manuscript is accepted for publication.

Latin names should be given in italics all over the manuscript.

The used abbreviation pKAL, sometimes written also as Pkal, is explained on the line 251 in the section of Materials and Methods. This abbreviation should be explained when it appears in the text for the first time. Moreover, the way how this abbreviation is written, should be unified.

The manuscript contains no real conclusion. Partial conclusive remarks can be found in the section of Discussion, however, a clearly stated Conclusion would improve the soudness of this manuscript.

When the corrections and addenda are made, then the manuscript will be ready for publication.

Author Response

Latin names should be given in italics all over the manuscript.

  • Thank you for your comments. According to your suggestion, we have corrected it.
  •  
  • The used abbreviation pKAL, sometimes written also as Pkal, is explained on the line 251 in the section of Materials and Methods. This abbreviation should be explained when it appears in the text for the first time. Moreover, the way how this abbreviation is written, should be unified..
  •  
  • Thank you for your comments. According to your suggestion, we have corrected it.
  •  
  • The manuscript contains no real conclusion. Partial conclusive remarks can be found in the section of Discussion, however, a clearly stated Conclusion would improve the soudness of this manuscript.
  •  
  • Thank you for your comments. According to your suggestion, we have corrected it.

    We corrected the conclusion clearly.

  • We used modest way to conclude the final results. We did not realized that this way may lead some audience misunderstand the findings. Thank you again for your comments.

Reviewer 2 Report

The manuscript deals with studies on the isolated polyphenols from Artemisia annua L exhibit anti-cancer effects on radio-resistant MDA-MB-231 4 human breast cancer cells and further investigating the related mechanisms by suppressing stem cell phenotype, β-catenin, and MMP-9. It seems providing the available results and explanation for anticancer polyphenols from A.annua. The paper may be accepted for publication except for very minor typewriting errors such as

“RT-R MDA-MB-231 cells” revised as “RT-R-MDA-MB-231 cells” in page 2, line 69, 71, and page 4, 129

Author Response

The paper may be accepted for publication except for very minor typewriting errors such as

“RT-R MDA-MB-231 cells” revised as “RT-R-MDA-MB-231 cells” in page 2, line 69, 71, and page 4, 129

  • Thank you for your comments. According to your suggestion, we have corrected it.